# Synthesis and Evaluation of Antifungal and Antibacterial Abilities of Carbon Nanotubes Grafted to Poly(2-hydroxyethyl methacrylate) Nanocomposites

**DOI:** 10.3390/polym15183657

**Published:** 2023-09-05

**Authors:** Karina Sandoval-García, Abraham G. Alvarado-Mendoza, Eulogio Orozco-Guareño, María A. Olea-Rodríguez, Leonardo R. Cajero-Zul, Sergio M. Nuño-Donlucas

**Affiliations:** 1Centro Universitario de Ciencias Exactas e Ingenierías, Universidad de Guadalajara, Guadalajara 44430, Mexico; karina.sandoval2995@alumnos.udg.mx; 2Departamento de Química, Centro Universitario de Ciencias Exactas e Ingenierías, Universidad de Guadalajara, Guadalajara 44430, Mexico; gabriel.alvarado@academicos.udg.mx (A.G.A.-M.); eulogio.orozco@academicos.udg.mx (E.O.-G.); 3Departamento de Farmacología, Centro Universitario de Ciencias Exactas e Ingenierías, Universidad de Guadalajara, Guadalajara 44430, Mexico; maria.olea@academicos.udg.mx; 4Departamento de Ingeniería Química, Centro Universitario de Ciencias Exactas e Ingenierías, Universidad de Guadalajara, Guadalajara 44430, Mexico; leonardo.cajero@academicos.udg.mx

**Keywords:** carbon nanotubes, nanocomposites, poly(2-hydroxyethyl methacrylate), antifungal capacity, antibacterial ability

## Abstract

Developing nanomaterials with the capacity to restrict the growth of bacteria and fungus is of current interest. In this study, nanocomposites of poly(2-hydroxyethyl methacrylate) (PHEMA) and carbon nanotubes (CNTs) functionalized with primary amine, hydroxyl, and carboxyl groups were prepared and characterized. An analysis by Fourier-transform infrared (FT-IR) spectroscopy showed that PHEMA chains were grafted to the functionalized CNTs. X-ray photoelectron spectroscopy suggested that the grafting reaction was viable. The morphology of the prepared nanocomposites studied by field-emission scanning electron microscopy (FE-SEM) and transmission electron microscopy (TEM) showed significant changes with respect to the observed for pure PHEMA. The thermal behavior of the nanocomposites studied by differential scanning calorimetry (DSC) revealed that the functionalized CNTs strongly affect the mobility of the PHEMA chains. Tests carried out by thermogravimetric analysis (TGA) were used to calculate the degree of grafting of the PHEMA chains. The ability of the prepared nanocomposites to inhibit the growth of the fungus *Candida albicans* and the bacteria *Staphylococcus aureus*, *Pseudomonas aeruginosa*, and *Escherichia coli* was evaluated. A reduced antifungal and antibacterial capacity of the prepared nanocomposites was determined.

## 1. Introduction

Among the various known biomaterials, poly(2-hydroxyethyl methacrylate) (PHEMA) stands out for its cytocompatibility, biocompatibility, and reduced immunological response from a host tissue [1]. PHEMA is a non-biodegradable, hydrophilic, stimuli-responsive, and optically transparent polymer, and it has been used extensively in the biomedical field. Its applications include hydrogel contact lenses [2], cancer therapy [3], bone tissue regeneration [4], wound healing [5], neural tissue engineering [6], and controlled drug delivery systems [7].

However, microbial contamination (which includes bacteria, fungi, and parasites) can limit the biomedical applications of PHEMA, because isolated PHEMA does not possess antimicrobial properties [1]. Isolated PHEMA cannot inhibit the growth of Gram-positive and Gram-negative bacteria. This drawback must be surpassed to increase the range of the biomedical potential uses of the PHEMA. Adopting effective antimicrobial strategies is necessary to achieve the safe use of polymer-based materials to make biomedical devices that improve human health. In this sense, the incorporation of antimicrobial drugs into PHEMA is a widely used approach [8,9]. However, implementing novel antimicrobial strategies is of current interest, because the prevalence of antibiotic-resistant bacterial infections prompts concerns [10].

Isolated PHEMA is rarely used. Typically, the precursor monomer (2-hydroxyethyl methacrylate) (HEMA) is copolymerized or the PHEMA is mixed with other materials to prepare a biomaterial of desirable properties. The selection of a material that is added to PHEMA plays a crucial role in the properties of a PHEMA-based prepared material. Nanostructured materials can be used as reinforcements of polymers at the nanoscale level [11]. Carbon nanotubes (CNTs) are materials with outstanding physical and chemical properties and have been used as reinforcement agents of polymers to prepare nanocomposites [12]. PHEMA-based nanocomposites reinforced with CNTs have been prepared and studied in previous studies. For instance, HEMA was polymerized on the surfaces of the multi-walled carbon nanotubes (MWCNTs) by free-radical polymerization using 2,2′azo bis(isobutyronitrile) (AIBN) as an initiator. The reaction product of this polymerization was studied by Fourier-transformed infrared (FT-IR) spectroscopy, and it was detected that chains of PHEMA were attached to the surface of MWCNTs [13]. In another study, PHEMA/MWCNT nanocomposites were prepared using the solvent casting method, specifically via the addition of PHEMA to a dispersion of MWCNTs in DMF. After the elimination of the solvent, a homogeneous dispersion of the MWCNTs was obtained. The prepared nanocomposites showed some improved mechanical properties compared to those of pure PHEMA and better thermal stability [14].

CNTs can act as antibacterial agents [15]. Unlike other nanomaterials, such as zinc oxide nanoparticles (ZnO NPs) which exhibit excellent antimicrobial properties [16], CNTs have at least two advantages that can modulate their antimicrobial action: (i) CNTs can be functionalized with chemical groups with bactericide properties [17]; (ii) the internal cavity of CNTs can be filled with a drug, whereby CNTs can act as drug reservoirs [18]. CNTs filled with an antimicrobial drug can be grafted to a biopolymer, and the nanocomposite obtained can be used in the drug delivery field [13].

CNTs have unique hexagonal structures and a high length-to-diameter ratio (for instance, up to 136,000,000:1) [19]. In addition, some of their properties (electrical, thermal, and mechanical) are better than those of other materials. However, the hydrophobic characteristics of the CNTs, and their ability to form strong van der Waals interactions, inhibit their dispersion in solvents and aqueous media. Because the development of novel antimicrobial materials with inhibitory bacterial capacity is an imperative necessity [20], research focused on the use of CNTs in the preparation of novel antibacterial materials is of current interest. Polymeric nanocomposites reinforced with CNTs can be used as antimicrobial agents. The main challenge to overcome for achieving the use of CNT-based nanocomposites as antimicrobial materials is to inhibit the natural self-aggregation of the CNTs, improve their dispersion, especially in water [17], and reduce their toxicity. In this sense, to prepare an effective antimicrobial polymeric nanomaterial containing CNTs, achieving the chemical functionalization of the CNTs is a desirable goal. Functionalized CNTs (f-CNTs) can be attached to polymeric chains reducing their natural tendency for self-aggregation and toxicity. Several types of chemical groups can decorate the surface of the CNTs, for instance, –NH_2_, which inhibits bacterial growth when present at high concentrations, while CNTs functionalized with –COOH and –OH groups have strong biocide properties (7 log reduction) against some pathogens [17,21]. Moreover, when a high dispersion driven by the formation of chemical bonds between f-CNTs and polymer chains is achieved, homogeneous nanomaterials are obtained, as reported elsewhere [22]. Homogeneous f-CNT-based nanomaterials with microbial inhibitory capacity are desirable materials in the nanomedicine field. Compared with other strategies used to prepare antibacterial nanocomposites based on the use of nanoparticles (such as gold, silver, platinum, and other oxide nanoparticles) [23], the high dispersion of f-CNTs chemically bonded to the polymeric chains of a polymer used as matrix can have a significant impact on the antibacterial capacity of the nanocomposite. This may be a consequence of the diffusion of the polymer chains bonded to f-CNTs in a medium containing microbes to produce a homogeneous dispersion of the f-CNTs.

The aim of this study was to synthesize and characterize nanocomposites of CNTs functionalized with primary amine, carboxyl, and hydroxyl groups and PHEMA, and to evaluate their antibacterial capacity. To the best of our knowledge, the synthetic chemical pathway used in this study to attach the prepared functionalized CNTs to the PHEMA chains has not been reported previously. The novelty of this study consists of determining the antimicrobial ability of our CNTs_amine_/PHEMA nanocomposites. The obtained nanocomposites were characterized by Fourier-transformed infrared (FT-IR) spectroscopy, X-ray photoelectron spectroscopy (XPS), field-emission scanning electron microscopy (FE-SEM), transmission electron microscopy (TEM), differential scanning calorimetry (DSC), and thermogravimetric analysis (TGA). In addition, the activity of isolated PHEMA and their nanocomposites against the growth of bacterial strains such as *Staphylococcus aureus* ATCC 6538, *Pseudomonas aeruginosa* ATCC 9027, *Escherichia coli* ATCC 25922, and the fungus *Candida albicans* ATCC 10231 was assessed.

## 2. Materials and Methods

### 2.1. Materials

Poly(2-hydroxyethyl methacrylate) (Mw = 20,000), trimethylamine (Et_3_N, 99%), oxalyl chloride (OxCl, 98%), dichloromethane (ACS reagent), ethylenediamine (EDA, >99%), and potassium bromide (KBr, FT-IR grade (>99%)) were provided by Sigma-Aldrich (Saint Louis, MO, USA). Alumina boats were acquired from Alfa Aesar (Tewksbury, MA, USA). Nitrogen gas (99.99%) was obtained from INFRA (Guadalajara, Mexico). Argon gas (99.998%) was obtained from PRAXAIR (Guadalajara, Mexico). Ferric nitrate nonahydrate (Fe(NO_3_)_3_·9H_2_O, hydrochloric acid (ACS reagent), and ethanol (99.9%) were purchased from Golden Bell (Guadalajara, Mexico). Deionized and distilled water was purchased from Productos Selectropura (Guadalajara, Mexico). Müller–Hinton agar was acquired from MCD LAB (Tlanepantla, Mexico). Trypticase in soy agar and in soy broth were obtained from BD Bioxon (Mexico City, Mexico). All of the chemical reagents were used without purification.

### 2.2. Synthesis, Purification, and Chemical Functionalization of the CNTs

CNTs were prepared via the CVD method using ethanol as a carbon source. For this, an experimental procedure previously used by our research group was followed, as reported elsewhere [24]. To achieve the purification of the prepared CNTs, a mixture of concentrated HNO_3_ (37.5 wt.%) diluted in water (1:3 *v*/*v*) was used. Additionally, the chemical functionalization with –NH_2_, –COOH, and –OH groups of the purified CNTs was carried out following an experimental procedure of three steps. Both procedures were developed and used by our research group previously, as reported elsewhere [25]. The experimental procedure used to functionalize the purified CNTs (named CNTs_pu_) is summarized below.

Dry CNTs_pu_ were partially oxidized. As a product of the oxidation process, hydroxyl and carboxyl groups were attached to the CNTs_pu_ surface. The contents of hydroxyl and carboxyl groups were calculated through two experimental procedures reported separately elsewhere [26,27]. Then, partially oxidized CNTs (named CNTs_ox_) reacted with OxCl with the goal of attaching acyl chloride functional groups onto the CNTs_ox_ surface. Using the product thus obtained (named CNTs_OCl_), a third step was performed. CNTs_OCl_ were reacted with EDA to attach amine groups to the ends of emerging side chains. The CNTs containing hydroxyl, carboxyl, and terminal amine groups (named CNTs_amine_) were used to prepare the PHEMA/CNTs_amine_ nanocomposites.

### 2.3. Synthesis of CNTs_amine_/PHEMA Nanocomposites

The PHEMA/CNTs_amine_ nanocomposites were synthesized following a procedure of four steps. Table 1 lists the names assigned to identify each nanocomposite, as well as the formulations used for the preparations. Two different contents of CNTs_amine_ (0.5 wt.% or 1.0 wt.%) were used in the syntheses. The description to prepare a PHEMA/CNTs_amine_ nanocomposite with 0.5 wt.% CNTs_amine_ is presented below.

Step 1. First, 1 g of PHEMA was placed into a 100-mL glass reactor. Then, 10 mL of dichloromethane was added. The dissolution was achieved by maintaining the mixture with constant stirring under a nitrogen atmosphere at room temperature (ca. 25 °C).

Step 2. The reactor was immersed in an ice bath at 0 °C. Then, 0.65 μL of Et_3_N was added. The mixture was stirred for 20 min. Then, 0.70 μL of OxCl dissolved in 5 mL of dichloromethane was added dropwise with a syringe. The mixture was allowed to react for 3 h. For this procedure, the hydroxyl groups of the PHEMA reacted with one acyl chloride group of OxCl, while the other acyl group was available to react in an additional step.

Step 3. Next, 0.05 g of CNTs_amine_ was added to the prepared mixture. Then, the reactor was immersed into oil bath at 30 °C. The dispersion was allowed to react for 24 h. After this, the product was introduced in an oven at 30 °C until a dry solid was obtained.

Step 4. The product of the reaction was purified by adding 40 mL of dichloromethane, and the mixture was mixed for 15 min. Next, the mixture was centrifuged at 7500 rpm. An insoluble solid (the PHEMA/CNTs_amine_ nanocomposite) was recovered. Step 4 was repeated twice. When a PHEMA/CNTs_amine_ nanocomposite was prepared with 1.0% CNTs_amine_, the amounts of the mentioned reagents were adjusted to maintain similar concentrations to those mentioned above. Figure 1 shows the chemical pathway used to synthesize the studied nanocomposites.

### 2.4. Characterization of CNTs_amine_/PHEMA Nanocomposites

A series of experimental techniques were employed to characterize the CNTs_amine_/PHEMA nanocomposites.

#### 2.4.1. Fourier-Transform Infrared Spectroscopy (FT-IR)

An analysis of the structure of pure PHEMA and the PHEMA/CNTs_amine_ nanocomposites was carried out using Fourier-transform infrared (FT-IR) spectroscopy with an FT-IR Spectrum One spectrophotometer (Perkin Elmer, Waltham, MA, USA). To perform the study, pellets of KBr and dry samples were prepared (at a ratio of 0.1 mg of the sample to 70 mg of KBr) by compression at room temperature (ca. 25 °C (298.15 K)). With the aim of reducing the signal/noise ratio, all reported spectra were recorded and analyzed from an average of 40 scans, with a resolution of 4 cm^−1^. The spectra of the analyzed samples were obtained at room temperature (ca. 25 °C (298.15 K)).

#### 2.4.2. X-ray Photoelectron Spectroscopy (XPS)

Pure PHEMA and the PHEMA/CNTs_amine_ nanocomposites were analyzed by X-ray photoelectron spectroscopy (XPS). The analysis was performed using equipment that comprised an XR 50 M monochromatic Al Kα_1_ (*hν* = 1486.7 eV) X-ray source and a Phoibos 150 spectrometer with a one-dimensional hemispheric detector 1D-DLD provided by SPECS (Berlin, Germany). The measurements were made at 150 W with an electron takeoff angle of 90°, a step size of 0.1 eV, and a pass energy of 10 eV. The base pressure was maintained at 4.2 × 10^−10^ mbar. The samples were mounted on a steel sample holder before performing the measurements using carbon tape. A flood gun was employed to compensate the charges on the sample. The obtained spectra were shifted according to C–C binding energy. The data obtained were fitted using the active background approach tool of the software Analyzer v. 1.42 [28].

#### 2.4.3. Field-Emission Scanning Electron Microscopy (FE-SEM)

The morphology of pure PHEMA and the PHEMA/CNTs_amine_ nanocomposites was analyzed by field-emission scanning electron microscopy (FE-SEM) using a MIRA 3LU microscope from Tescan (Brno, Czech Republic). Samples were previously dried in an oven at 60 °C (333.15 K) for 72 h. Then, ca. 0.01 g of sample was mixed with 1 cm^3^ of acetone at room temperature (ca. 25 °C (298.15 K)). The obtained mixture was sonicated for 5 min, and an aliquot of the prepared dispersion was taken with a Pasteur pipette and poured onto a Cu grid. The solvent was evaporated completely. Then, the samples were introduced to a SCD004 golden evaporator from BalTec AG (Pfäffikon, Switzerland), and a gold layer was applied onto the sample surface via a 20-s electrodeposition process. Afterward, the sample were examined in an FE-SEM microscope.

#### 2.4.4. Transmission Electron Microscopy (TEM)

Samples of pure PHEMA and the PHEMA/CNTs_amine_ nanocomposites were examined using a 1010 TEM from JEOL (Peabody, MA, USA) operated at 200 kV. Before the analysis, the samples were dried in an oven at 60 °C (333.15 K) for 72 h. Then, approximately 0.010 g of each sample was mixed with 2 mL of acetone at room temperature. Afterward, the mixture was sonicated for 5 min, and, using a Pasteur pipette, an aliquot of the prepared dispersion was poured onto Cu grid. The solvent was evaporated through action of a 60 W solar lamp for 20 min. Then, the sample thus obtained was analyzed.

#### 2.4.5. Differential Scanning Calorimetry (DSC)

Thermal characterization of pure PHEMA and the PHEMA/CNTs_amine_ nanocomposites was performed by differential scanning calorimetry (DSC) using a Q100 calorimeter from TA Instruments (New Castle, DE, USA). The mass of samples was in the range from 3 to 5 mg. DSC thermograms were obtained following a heating program from −80 °C (193.15 K) to 200 °C (473.15 K) at a heating rate of 10 K/min using a nitrogen flow rate of 50 cm^3^/min. All tests were performed while maintaining an inert atmosphere. Two heating scans were recorded, and the second scan is reported.

#### 2.4.6. Thermogravimetric Analysis (TGA)

Thermal analysis of pure PHEMA and the PHEMA/CNTs_amine_ nanocomposites was completed using thermogravimetric analysis (TGA). The measurements were recorded in a TGA5000 Discovery thermobalance from TA Instruments (New Castle, DE, USA) operated in the range from 25 °C (298.15 K) to 600 °C (873.15 K) at 10 K/min under a constant nitrogen flow rate of 25 mL/min to create an inert atmosphere. The mass of the samples ranged from 3 to 10 mg.

#### 2.4.7. Evaluation of the Antibacterial and Antifungal Abilities of Pure PHEMA and the CNTs_amine_/PHEMA Nanocomposites

The inhibitory ability of the pure PHEMA and the PHEMA/CNTs_amine_ nanocomposites against bacterial strains of *Staphylococcus aureus* ATCC 6538, *Pseudomonas aeruginosa* ATCC 9027, and *Escherichia coli* ATCC 25922 was studied. Moreover, their ability to inhibit the growth of the fungus *Candida albicans* ATCC 10231 was evaluated. The experimental procedure is described next. First, the fungus and each type of bacteria were reactivated in trypticase in soy broth with yeast extract (CSTEL) and incubated at 35 ± 2 °C (308.15 ± 2 K) for 24 h, separately. Bacteria and fungi were obtained via two consecutive reseedings. Second, each bacterium or fungus was inoculated on Petri dishes with Müller–Hinton agar. Then, the selected samples were added. The halo of inhibition was assessed after incubation at 35 °C (308.15 K) for 24 h [29].

## 3. Results and Discussion

As previously reported in a study carried out by our research group, the surface of the CNTs_amine_ was decorated with primary amine, hydroxyl, and carboxyl groups [25]. The contents of hydroxyl and carboxyl groups were 13.3 wt.%, and 37.5 wt.% per gram of the CNTs_ox_ (precursor of the CNTs_amine_), respectively, while the presence of the primary amine groups was documented through an analysis using FT-IR and XPS spectroscopies.

### 3.1. Analysis of the Chemical Structure of CNTs_amine_/PHEMA Nanocomposites

Fourier-transform infrared (FT-IR) spectroscopy is an adequate technique to analyze chemical groups inserted on the walls of CNTs. FT-IR spectra of the pure PHEMA (Figure 1A), Na 1 (Figure 1B), Na 2 (Figure 1C), Na 3 (Figure 1D), and Na 4 (Figure 1E) are shown in Figure 1. In the FT-IR spectrum of pure PHEMA (Figure 1A), typical spectral contributions of this polymer could be detected, as reported elsewhere [30]. Accordingly, the most intense band was detected at 1726 cm^−1^, attributed to the stretching vibration of the carbonyl bond of an ester group. The stretching vibration of the hydroxyl group produced the wide and intense band detected at 3400 cm^−1^. Additionally, the band observed at 1160 cm^−1^ was caused by torsional vibration of the C–H bond and symmetric stretching vibration of the C–O–C functionality of the ester group. The FT-IR spectra of the nanocomposites showed similar spectral behavior. In these spectra, significant changes with respect to those observed in the spectrum of pure PHEMA were detected. Furthermore, additional bands were observed. Thus, the weak band at 1630 cm^−1^ was attributed to the amide II band due to flexion of the –NH_2_ functionality of the amide group. Moreover, the band assigned to the carbonyl bond of an ester group was detected at 1733 cm^−1^. As a shoulder of this band, a new band was detected at 1773 cm^−1^, attributed to asymmetric stretching vibration of the two carbonyl functionalities of the anhydride group. In Figure 2, partial FT-IR spectra of pure PHEMA, Na 2, and Na 3 are shown. Arrows are used to identify the spectral contributions detected at 1733 cm^−1^ and 1773 cm^−1^ in the spectra of Na 2 and Na 3. The formation of amide and anhydride groups indicates that chemical reactions were carried out between acyl chloride groups added to the PHEMA and the primary amine or carboxyl groups inserted on the walls of the CNTs_amine_. Through the amide and anhydride groups, the chains of PHEMA were grafted to the walls of CNTs_amine_ as shown in Figure 1. It is worth mentioning that, in Figure 1, there is evidence of the presence of residual chemical groups of the hydroxyl, carboxyl, and C–Cl functionalities. Thus, a wide band appeared at 3440 cm^−1^ due to the stretching vibrations of hydroxyl groups. A wide band due to stretching vibration of the O–H bond of carboxyl functionality was detected at 2603 cm^−1^. Moreover, a band caused by the stretching vibration of the C–Cl bond could be observed at 806 cm^−1^. Table 2 lists the above-described FT-IR bands of pure PHEMA and Na 3.

The chemical composition and the electronic structure of PHEMA and the PHEMA/CNTs_amine_ nanocomposites were studied by X-ray photoelectron spectroscopy (XPS). The analysis by XPS contributed to obtaining additional knowledge of the structure of the prepared nanocomposites. In the Appendix A, the survey spectra of pure PHEMA and of all PHEMA/CNTs_amine_ nanocomposites are shown. Figure 3 shows the *C1s* core-level normalized spectra of PHEMA and Na 1. The presented XPS spectra were adjusted to the respective shift taking as reference the signal of C–C/C–H bonds at 285.0 eV. The *C1s* core-level normalized spectrum of pure PHEMA (Figure 3A) showed components at five positions. The components and their atomic concentrations were C–C/C–H bonds at 285.0 eV [31] (21.1%), C–C=O bonds at 285.7 eV [31] (15.2%), C–OH bonds at 286.7 eV [31] (14.6%), C–O–C bonds at 287.2 eV [31] (6.1%), and O–C=O bonds at 289.2 eV [31] (10.5%). Moreover, the *C1s* core-level normalized spectrum of Na 1 also showed components at five positions. These components and their atomic concentrations were C–C/C–H bonds at 285.0 eV [31] (29.8%), C–C=O bonds at 285.7 eV [31] (14.6%), C–OH bonds at 286.7 eV [31] (8.0%), C–O–C [25]/HN–C=O bonds [26,27,32,33] at 287.2 eV (7.2%), and O–C=O bonds at 289.3 eV [31] (6.9%). It is evident that the atomic concentration of the C–OH bonds detected in the spectrum of Na 1 decreased with respect to that observed in the spectrum of pure PHEMA. This fact suggests that this functionality reacted. Specifically, the reaction was performed between the hydroxyl groups of PHEMA and the acyl chloride groups of OxCl. In a similar fashion, the slight increment in atomic concentration attributed to C–O–C/HN–C=O bonds detected in the spectrum of Na 1 with respect to those observed in the spectrum of pure PHEMA strongly suggests that another chemical reaction took place. Now, the well-known chemical reaction between acyl chloride groups and primary amine groups to produce amide groups was carried out. This analysis is in accordance with the described FT-IR spectra shown in Figure 1 and Figure 2, and supports the chemical path presented in Figure 1.

Figure 4 shows the *O1s* core-level normalized spectra of PHEMA and Na 1. In the case of the PHEMA (Figure 4A), the *O1s* XPS spectrum showed components at three positions. The components and their atomic concentrations were O=C/O–C=O bonds at 532.7 eV [31] (13.8%), HO–C bonds at 533.7 eV [31] (12.6%), and C–O–C bonds at 534.4 eV [31] (5.2%). Similarly, the *O1s* XPS spectrum of the Na 1 also showed components at three positions. The components and their atomic concentrations were O=C/O–C=O bonds at 532.7 eV [31] (14.4%), HO–C bonds at 533.8 eV [31] (6.9%), and C–O–C bonds at 534.5 eV [31] (3.1%). The most significant change between these spectra was a slight increment in the atomic composition of the O=C/O–C=O bonds detected in the *O1s* XPS spectrum of Na 1 compared to that detected in the PHEMA. This increment was attributed to a chemical reaction developed between acyl chloride and carboxyl groups to produce ester groups, as presented in Figure 1.

The XPS results suggest that the carbon nanotubes functionalized with hydroxyl, carboxyl, and primary amine groups were chemically attached to the polymer chains of PHEMA through amide and ester groups.

### 3.2. Morphological Analysis of CNTs_amine_/PHEMA Nanocomposites

Figure 5 shows the FE-SEM micrographs of pure PHEMA (Figure 5A) and Na 4 (Figure 5B,C). The micrograph of pure PHEMA showed a smooth surface with sharp cuts at one end. These cuts were a consequence of the brittle nature of the PHEMA, whose glass transition temperature (T_g_) is higher than the room temperature (ca. 298.15 K) at which the FE-SEM analysis was performed. On the contrary, the morphology of Na 4 showed significant changes, whereby an irregular morphology was observed (Figure 5B). There was an influence of the CNTs_amine_ used for the preparation of the studied nanocomposite on the change in morphology detected, as observed in Figure 5C where a CNT emerging from a globular structure detected on the surface could be observed. An arrow points to the CNT. These results strongly suggest that CNTs_amine_ were grafted to the PHEMA chains. In this sense, in a previous report [34], a significant change in the morphology of PHEMA-based nanocomposites with respect to the observed nanofiller was explained as being due to the grafting of halloysite nanotubes to the polymeric chains of PHEMA. Similarly, in another study, an analysis of FE-SEM micrographs of nanocomposites of Fe_3_O_4_ nanoparticles/PHEMA showed a drastic change with respect to those of pure PHEMA. This was explained as being due to the adhesion of Fe_3_O_4_ nanoparticles to the PHEMA chains [35]. The Appendix A show the FE-SEM micrographs of Na 1, Na 2, and Na 3.

Figure 6 depicts the TEM micrographs of pure PHEMA (Figure 6A) and Na 4 (Figure 6B). The TEM micrograph of pure PHEMA showed darker domains, indicating the overlapping of layers of PHEMA [36]. Moreover, circular regions could be observed which were probably formed during the evaporation of the solvent used to prepare the solution of PHEMA for PHEMA film formation. The edges of these regions were also darker due to the agglomeration of polymeric layers. The TEM micrograph of Na 4 showed the same darker domains, but there was also the presence of tubular structures, which were attributed to the CNTs used for nanocomposite preparation. The length of these tubular structures was on the order of microns. This length was typical of the CNTs, highlighting a CNT that extended over the entire surface examined. The described morphology of Na 4 indicates that the CNTs_amine_ were dispersed in the polymeric matrix of PHEMA.

### 3.3. Thermal Analysis of CNTs_amine_/PHEMA Nanocomposites

Figure 7 shows the DSC thermograms of pure PHEMA (Figure 7A) and their nanocomposites (Figure 7B–E). PHEMA is an amorphous polymer. The DSC thermograms showed only one thermal event: glass transition relaxation. The measured glass transition temperature (T_g_) of PHEMA was 75 °C (348.15 K). For their nanocomposites, it is evident that the intensity of the change in the baseline of each thermogram caused by the glass transition relaxation decreased. Table 3 lists the glass transition temperatures of all the materials studied. The glass transition temperatures of the nanocomposites were higher than that of pure PHEMA. As expected, as the content of CNTs_amine_ used in the nanocomposite preparation increased, T_g_ of the prepared nanocomposites increased. The measured T_g_ of Na 4 was 371.15 K (the highest of all the nanocomposites studied), being 23 K higher than that of pure PHEMA. The thermal behavior described could be explained by the reduction in the mobility of the polymeric chains of PHEMA caused by the chemical bonds formed with CNTs_amine_. In this sense, this nanofiller acted as an agent that produced a constraint effect on the dynamics of the polymer chains of PHEMA. In a previous study by our research group, similar results were found for nanocomposites prepared with polyethylene glycol and CNTs functionalized with amine groups [25].

Figure 8 shows the TGA thermograms of pure PHEMA and Na 2. The TGA thermograms of the other studied nanocomposites were very similar to that of Na 2. For pure PHEMA, a small thermal decomposition was detected at temperatures below 265 °C (538.15 K), which could mainly be attributed to water loss. The main weight loss was observed in the range from ca. 265 °C (538.15 K) to 500 °C (773.15 K) due to the decomposition of the polymer chains of PHEMA. For Na 2. the thermal decomposition was developed in three stages: (i) the first from ca. 90 °C (363.15 K) to 265 °C (538.15 K) followed by a temperature range (from 265 °C (538.15 K) to 275 °C (548.15 K), in which a small weight loss was recorded; (ii) the second from 275 °C (548.15 K) to 490 °C (763.15 K); (iii) the third from 490 °C (763.15 K) to 800 °C (1073.15 K). The weight loss of the first stage was 58.7%, while that for the second stage was 31.2%. The weight loss of the first stage was attributed to degradation of the chemical groups attached to the surface of CNTs_amine_ not linked to PHEMA chains, and the weight loss of the second stage was attributed to the degradation of the PHEMA chains chemically linked to the CNTs_amine_. In a previous study, a similar thermal decomposition behavior was described for nanocomposites of PHEMA chains grafted to sidewalls of multi-walled carbon nanotubes (MWCNTs) prepared by fragmentation chain transfer (RAFT) polymerization [37]. From the weight loss of the first and the second stages, the degree of PHEMA chains grafted to the CNTs in Na 2 was calculated at 36.7%. Table 3 lists the grafting degrees of the nanocomposites studied. Na 4 presented the highest grafting degree of all analyzed nanocomposites. This result is in accordance with the large amount of CNTs_amine_ (1 wt.%) used in its preparation. Due to the higher amount of CNTs_amine_ used to prepare the nanocomposites, there were more chemical groups available to develop the grafting reaction with the PHEMA chains. The preparation of nanocomposites with polymer chains grafted to functionalized CNTs improved the dispersion of this nanofiller, which is a critical issue to obtain antimicrobial nanocomposites with desirable performance.

### 3.4. Evaluation of the Antifungal and Antibacterial Abilities of the CNTs_amine_/PHEMA Nanocomposites

Figure 9 shows four photographs of sections of Petri dishes where Na 4 samples were inoculated with the fungus *Candida albicans* (A) or the bacteria *Pseudomona aeruginosa* (B), *Escherichia coli* (C), and *Staphylococcus aureus* (D). Photographs of the complete Petri dishes of antibacterial tests of all nanocomposites are presented in the Appendix A. The photographs presented in Figure 9 evidence an inhibition halo surrounding each sample of Na 4. This halo took a colored appearance and marked the region where diffusion of the chains of PHEMA grafted to CNTs_amine_ inhibited the growth of the fungus or the bacteria studied. The diameters of the observed halo for all nanocomposites studied are shown in Table 4. The diameter of the halo produced for the action of Na 3 was larger than that observed in the tests with the other nanocomposites. These results indicate that the studied nanocomposites had antifungal and antibacterial abilities. However, these abilities were reduced and dependent on the content of the CNTs_amine_ used in their preparation. For Na 1 and Na 2 (both prepared with 0.5% CNTs_amine_), this ability was small or non-existent. On the contrary, for Na 3 and Na 4 (prepared with 1.0% CNTs_amine_), the inhibitory capacity was improved. The well-known antimicrobial capacity of CNTs is related to their ability to act as needles that pierce the cellular membrane, contributing to the disruption of RNA or DNA replication, which causes cell death [38]. According to the experimental evidence compiled from this study, two factors contribute to the potential mechanism of inhibition that can help in explaining the bactericidal ability of the nanocomposites studied. Firstly, the carboxyl and hydroxyl functionalities that decorate the surface of the CNTs_amine_ have strong bactericide abilities [17]. This ability could be reinforced by the fact that the functionalized CNTs can bind to microbial cells easily, as reported elsewhere [39]. Secondly, the effect of an increased amount of CNTs_amine_ on the antimicrobial ability of the nanocomposites, was detected as an increase in the inhibition diameter size. A similar concentration-dependent effect was previously reported [40].

The reduced antimicrobial capacity of CNTs_amine_/PHEMA nanocomposites was lower compared with the nanocomposites prepared with CNTs and a biopolymer with antimicrobial properties. For example, chitosan is a biodegradable and biocompatible biopolymer with antimicrobial properties [41]. Nanocomposites of CNTs–chitosan showed a synergistic effect on antimicrobial properties [42]. In this sense, CNTs functionalized with hydroxyl and carboxyl chemical groups were used to prepare chitosan-based nanocomposites. These nanocomposites were prepared dispersing different amounts of functionalized CNTs in a solution of chitosan prepared with 40 mg of chitosan dissolved in 40 mL of 0.1 M aqueous acetic acid. The nanocomposite prepared with 50 mg of functionalized CNTs produced inhibition halos of 18.20 ± 0.05 mm for *S. aureus*, 14.50 ± 0.08 mm for *E. coli*, and 19.20 ± 0.10 mm for *P. aeruginosa* [43]. A simple comparison of these results with the results of our study highlights the relevance of the polymer chosen as the matrix of antimicrobial nanocomposites containing functionalized CNTs.

## 4. Conclusions

The preparation of CNTs_amine_/PHEMA nanocomposites was carried out successfully. CNTs_amine_ was obtained via functionalization of purified CNTs. FT-IR spectroscopy analysis revealed that CNTs_amine_ were linked through amide and ester groups to PHEMA chains, while XPS spectroscopy analysis suggested that the grafting reaction was possible. Micrographs of the prepared nanocomposites obtained by FE-SEM and TEM indicated that CNTs_amine_ were immersed in the PHEMA polymer matrix without apparent segregation. An increment in the T_g_s of the prepared nanocomposites with respect to the T_g_ of pure PHEMA was detected by DSC. This increment was produced via a constraint effect induced by the CNTs_amine_ attached to the PHEMA chains. The grafting degree of PHEMA chains on the CNTs_amine_ was determined by TGA. In general, this degree increased with the content of CNTs_amine_ in the prepared nanocomposites. The CNTs_amine_/PHEMA nanocomposites showed reduced antifungal and antibacterial abilities against the fungus *C. albicans* and the bacteria *P. aeruginosa*, *E. coli*, and *S. aureus*.

## Data Availability

The data presented in this study are available on request from the corresponding author.

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
