# Peer review of "Synthesis and Evaluation of Antifungal and Antibacterial Abilities of Carbon Nanotubes Grafted to Poly(2-hydroxyethyl methacrylate) Nanocomposites"

_polymers, 2023, doi:10.3390/polym15183657_

Round 1

Reviewer 1 Report

The paper presents interesting and new enough results to be accepted for publication after major revision. The following issues should be clarified.

The authors noted "Nanostructured materials can be used as nanoscale-level reinforcements of polymers. Carbon nanotubes (CNTs) are materials with outstanding physical and chemical properties and had been used as reinforcement agents of polymers to prepare nanocomposites.", however, references are absent in these placed 

The authors suggest "Several types of chemical groups can decorate the surface of the CNTs, for instance, cationic group (as –NH2), which inhibits bacterial growth when they have present at high concentrations, while CNTs functionalized with anionic (–COOH)....." This sentence should be corrected. The cationic group is -NH3, anionic -COO-.

The quality of Figure 1 is low. Adding a table that includes the wavenumbers of the peaks and their corresponding assigned chemical groups would be a valuable addition to the manuscript. This table will provide readers with a concise and easily accessible reference to understand the identified chemical functionalities and their corresponding spectral locations. It will enhance the clarity of the presented data and facilitate further interpretation of the results. 

Authors noted, "The XPS results indicate that the functionalized carbon nanotubes with hydroxyl, carboxyl, and primary amine groups are chemical attached to the polymer chains of PHEMA through amide and ester groups" This assertion should be explained in the details because including the functionalized nanotubes in PHEMA leads to a change in the proportion of the functional groups, especially C-OH bonds. Additionally, including the functionalized nanotubes which have amide and ester groups is not an endorsement of the formation of the new amide and ester bonds. 

The results of the TGA are completely not clear to me. It is a well-known fact that thermal degradation of the carbon nanotubes even oxidized is at the highest temperatures than polymers; in Figure 7  degradations of the pure PHEMA and PHEMA composite were entirely finalized at the same temperatures.

It will be valuable to add information on the content of the CNTs used in the nanocomposite preparation to Table 2. 

The PHEMA composites with functionalized nanotubes demonstrated relatively low antimicrobial properties. The authors' optimism regarding the expected excellent results in this context remains unclear. Therefore, a thorough discussion of the potential mechanisms of antimicrobial action should be included. Furthermore, it would be valuable to incorporate a Table that presents inhibition zones to provide a concise summary of the antimicrobial efficacy.

The English should be carefully checked. Using scientific jargon is unacceptable.  For example, young bacteria.

Moderate editing of English language required.

Author Response

The paper presents interesting and new enough results to be accepted for publication after major revision. The following issues should be clarified.

The authors noted "Nanostructured materials can be used as nanoscale-level reinforcements of polymers. Carbon nanotubes (CNTs) are materials with outstanding physical and chemical properties and had been used as reinforcement agents of polymers to prepare nanocomposites.", however, references are absent in these placed 

We agree with the comment of the reviewer. In the revised manuscript two references were added to support the statement indicated.

The authors suggest "Several types of chemical groups can decorate the surface of the CNTs, for instance, cationic group (as –NH2), which inhibits bacterial growth when they have present at high concentrations, while CNTs functionalized with anionic (–COOH)....." This sentence should be corrected. The cationic group is -NH3, anionic -COO-.

We agree with the comment of the reviewer. In the revised manuscript these mistakes were corrected eliminating the categorizations cationic, anionic, and neutral. However, the mention of the chemical groups (-NH2, -COOH, and -OH) used to functionalize the CNTs was left unchanged, because these chemical groups have antimicrobial capacity and their mention contributes to explaining the potential mechanism of the antimicrobial ability of the CNTsamine/PHEMA nanocomposites.

The quality of Figure 1 is low. Adding a table that includes the wavenumbers of the peaks and their corresponding assigned chemical groups would be a valuable addition to the manuscript. This table will provide readers with a concise and easily accessible reference to understand the identified chemical functionalities and their corresponding spectral locations. It will enhance the clarity of the presented data and facilitate further interpretation of the results.

We agree with the reviewer. In the revised manuscript, a new Table that shows and describes the spectral contributions observed in Figure 1 was added. Also, Figure 1 was modified and the FT-IR spectra of Na 1 and Na 4 were included. Additionally, a new Figure in which partial FT-IR spectra of pure PHEMA, Na 2, and Na 3 in the region from 1400 cm-1 to 2000 cm-1, was presented. This new Figure improves the clarity of the original manuscript.

Authors noted, "The XPS results indicate that the functionalized carbon nanotubes with hydroxyl, carboxyl, and primary amine groups are chemical attached to the polymer chains of PHEMA through amide and ester groups" This assertion should be explained in the details because including the functionalized nanotubes in PHEMA leads to a change in the proportion of the functional groups, especially C-OH bonds. Additionally, including the functionalized nanotubes which have amide and ester groups is not an endorsement of the formation of the new amide and ester bonds. 

We agree with the reviewer. The mentioned sentence above was modified in the revised manuscript. The XPS technique can only indicate the viability of the chemical path shown in the Scheme I. Also, the abstract and conclusions were modified.

The results of the TGA are completely not clear to me. It is a well-known fact that thermal degradation of the carbon nanotubes even oxidized is at the highest temperatures than polymers; in Figure 7 degradations of the pure PHEMA and PHEMA composite were entirely finalized at the same temperatures.

We do not completely agree with these reviewer´s observations. Carbon nanotubes maintain their thermal stability at high temperatures, higher than the observed for polymers, indeed. Our TGA results compared the thermal degradation behavior of pure PHEMA and Na 2. The Na 2 was prepared with a 0.5 wt.% content of CNTsamine. Therefore, the high content of PHEMA determined the general thermal stability of this sample. In the original manuscript, it was mentioned that the TGA curve of Na 2 (in a similar way to the observed in the TGA curves of all the studied nanocomposites) presented three stages. This behavior is different from the observed in the TGA curve of the pure PHEMA which presented one degradation stage only, and for which the degradation was completed at 470 ºC. It is noticeable to observe that at temperatures higher than 480 ºC, the TGA curve of Na 2 maintained a small mass, and the thermal degradation was not completed yet. We consider that this thermal stability can be explained by the presence of the CNTsamine between the PHEMA chains. As the temperature increased, the PHEMA chains lost their stability, and the structure of Na 2 degraded completely. It is possible that the low mass of CNTsamine remained unchanged at the final of the test, but the equipment did not detect it.

It will be valuable to add information on the content of the CNTs used in the nanocomposite preparation to Table 2. 

The content of CNTsamine used to prepare the nanocomposites was shown in Table 1. Also, the name of the nanocomposites was added in Table 1. In the original manuscript, the name of the nanocomposites was presented in Table 2. We believe that repeating the content of CNTsamine is not needed in order to increase the clarity of the manuscript. However, in the revised manuscript the legend ‘Content of CNTsamine’ was explicitly stated in Table 1.

The PHEMA composites with functionalized nanotubes demonstrated relatively low antimicrobial properties. The authors' optimism regarding the expected excellent results in this context remains unclear. Therefore, a thorough discussion of the potential mechanisms of antimicrobial action should be included. Furthermore, it would be valuable to incorporate a Table that presents inhibition zones to provide a concise summary of the antimicrobial efficacy.

In the revised manuscript a new Table (called Table 4) in which the size of the diameter of the halo of inhibition of pure PHEMA and all CNTsamine/PHEMA nanocomposites was added. The nanocomposites prepared with higher content of CNTsamine (1.0 wt.%) showed better inhibition ability. This means that the content of CNTsamine played a crucial role to achieve the death of bacteria and fungus. Moreover, an additional explanation of our results was presented discussing the factors that contributed to the potential mechanisms of the antimicrobial action of the studied nanocomposites.

The English should be carefully checked. Using scientific jargon is unacceptable.  For example, young bacteria.

In the revised manuscript the English was checked and improved. The expression ‘Young bacteria’ was eliminated.

Reviewer 2 Report

Comments

The comments for the article, ‘Synthesis and Evaluation of Antifungal and Antibacterial Abilities of Carbon Nanotubes Grafted to Poly(2-hydroxyethyl 4 methacrylate) Nanocomposites’ are given below,

·         The authors kept the same figures for 1b and 1c mistakenly. And also include the IR spectrum of Na1 and Na4 along with it.

·         The XPS spectra is confusing. C1s shows five peaks altogether and O1s shows three peaks altogether. The deconvoluted peaks of C1s indicates the presence of 4 bonding between C and O and 1 bonding between C-C/C-H. If this is so, then O1s should also have 4 deconvoluted peaks. But there are only three deconvoluted peaks for O1s. Explanation is needed in this regards. It will be better to include the survey spectrum of all materials (PHEMA, Na1, Na2, Na3 and Na4) for clear understanding.

·         In the SEM images, the tubular structure of CNT is not seen. Include the SEM images of all samples ((PHEMA, Na1, Na2, Na3 and Na4) at different magnifications (fix to similar magnifications for all samples for comparison) to properly examine the morphology.

·         TEM images are with different magnifications for both the samples. Generally comparison should be made with similar magnifications. As it is TEM particles with nano (up to 100 nm) size are given much importance. Take TEM images at higher magnifications (0 – 100 nm), make sure to take similar magnification for comparison.

·         In the DSC thermogram, Fig. 6E do not show any transitions, only a straight line is observed. Account for it.

·         Figure 8, representing the antibacterial and antifungal images in this way is not a proper way. Include the images of the entire petridish. Only image of some part of the petridish is given. What about the antibacterial/fungal property of the other materials (PHEMA, Na1, Na2, Na3).

Author Response

The comments for the article, ‘Synthesis and Evaluation of Antifungal and Antibacterial Abilities of Carbon Nanotubes Grafted to Poly(2-hydroxyethyl 4 methacrylate) Nanocomposites’ are given below,

  • The authors kept the same figures for 1b and 1c mistakenly. And also include the IR spectrum of Na1 and Na4 along with it.

The IR spectra shown in Figures 1B and 1C are different. A careful comparison of the spectrum presented in Figure 1B and the spectrum of Figure 1C makes it possible to distinguish that there are many subtle differences between both spectra. For example, the shoulder of the band observed at 1733 cm-1 was detected at 1773 cm-1 and appears more defined in the spectrum of Figure 1C with respect to the one observed in Figure 1B. Moreover, in the spectrum of Na 3 (Fig. 1C) a very weak band was detected at 2850 cm-1 which was not observed in the spectrum of Na 2 (Fig. 1B). Other differences can be detected, and not mentioned here. In the revised manuscript, Figure 1 was re-edited and now the spectrum of all nanocomposites and the spectrum of pure PHEMA are presented. A new Figure, called Figure 2 in which partial IR spectra of pure PHEMA, Na 2, and Na 3 in the region from 1400 cm-1 to 2000 cm-1, was presented. The aim of this addition is to improve the clarity of the IR analysis of studied nanocomposites. Also, a new Table listing the spectral contributions detected is presented.

  • The XPS spectra is confusing. C1s shows five peaks altogether and O1s shows three peaks altogether. The deconvoluted peaks of C1s indicates the presence of 4 bonding between C and O and 1 bonding between C-C/C-H. If this is so, then O1s should also have 4 deconvoluted peaks. But there are only three deconvoluted peaks for O1s. Explanation is needed in this regards. It will be better to include the survey spectrum of all materials (PHEMA, Na1, Na2, Na3 and Na4) for clear understanding.

The Supplementary Material of the revised manuscript shows the survey spectrum of PHEMA and all nanocomposites. Regarding the confusion mentioned by the reviewer, the explanation is presented next. Indeed, in the O1s spectra the three deconvoluted peaks were detected, while in the C1s five peaks can be observed, three out of them are related with C and O bonds. The peak detected at 285.7 eV was assigned to the C–C=O bonds. This means that this assignation is not a carbon-oxygen bond. In addition, the peak at 285 eV was assigned to C–C/C–H bonds. In the images below, are demonstrated the assignations and the chemical structure of PHEMA as it is reported in the literature and in which we support our analysis ‘Briggs, G. Beamson - High Resolution XPS of Organic Polymers_ The Scienta ESCA300 Database-Wiley (1992)’.

  • In the SEM images, the tubular structure of CNT is not seen. Include the SEM images of all samples ((PHEMA, Na1, Na2, Na3 and Na4) at different magnifications (fix to similar magnifications for all samples for comparison) to properly examine the morphology.

In the revised manuscript, an arrow was added in Figure 5C to indicate a structure tubular that emerges from the polymeric surface of the Na 4 sample. This structure resembles a CNT. Therefore, in the revised manuscript an additional sentence was written to indicate that the mentioned structure was attributed to a CNT. In the FE-SEM micrographs of the pure PHEMA (as the one shown in Figure 5) it was not observed a structure tubular in the surface examined. On the contrary, a tubular structure also was observed in a FE-SEM micrograph of Na 2. Although we agree with the comment of the reviewer, at present we do not have micrographs available of the PHEMA and of Na 4 with the same magnification that show a tubular structure as was described here. As a complement of the Figure 5, in the Supplementary Material are shown FE-SEM micrographs of Na 1, Na 2, and Na 3.

  • TEM images are with different magnifications for both the samples. Generally comparison should be made with similar magnifications. As it is TEM particles with nano (up to 100 nm) size are given much importance. Take TEM images at higher magnifications (0 – 100 nm), make sure to take similar magnification for comparison.

In the revised manuscript are presented two TEM micrographs with the same magnification (20 kx) and equal bar scale (1000 nm). One of them corresponds to the TEM micrograph of the pure PHEMA presented in the original manuscript, while the other is a new TEM micrograph of the Na 4. In the micrograph of Na 4 a tubular structure attributed to a CNT was detected, which crosses all observed surface. In the revised manuscript a new sentence is presented to describe the observed structure. Regarding the particles observed with nano size, at present we do not have new available TEM micrographs that provide more information about its characteristics. It is important of mention that the aim of the TEM analysis was to find evidence of the presence of CNTs on the surface of the nanocomposites examined.

  • In the DSC thermogram, Fig. 6E do not show any transitions, only a straight line is observed. Account for it.

We do not completely agree with this reviewer´s observation. Indeed, the DSC thermogram of Na 3 is basically resolved as a straight line. This thermogram is shown in the revised manuscript in the Figure 7E. However, when the thermogram is analyzed with the software of the calorimeter used (model Q100 of TA Instrument) a very weak change in the base line associated with a typical change produced by a glass transition was detected. Unfortunately, in this moment we do not have other equipment available to support the measurement of the Tg of the Na 3.

  • Figure 8, representing the antibacterial and antifungal images in this way is not a proper way. Include the images of the entire petridish. Only image of some part of the petridish is given. What about the antibacterial/fungal property of the other materials (PHEMA, Na1, Na2, Na3).

In the revised manuscript it was added a new table (called Table 4), in which is reported the diameter of inhibition of all nanocomposites studied. Regarding the images of partial Petri dishes, this presentation was carried out to concentrate all the information in one Figure only. We agree with the reviewer. Therefore, new pictures of complete Petri dishes of all nanocomposites studied were included in the Supplementary Material.

Reviewer 3 Report

The manuscript shows good quality and is well written, supported by their data. However, the authors should consider the following:

1.    Introduction should also include why use CNT as filler, what advantages compared to other nanoparticles like ZnO.

2.    Maybe I’m missing, could author indicate what is the size of CNT like length and diameter. It will be better if the author can determine TEM  of CNT and f-CNT itself.

3.    The author should compare the impact of their studies against other toughening strategies in the literature

4.    The originality and novelties of the paper need to be further clarified.

Author Response

The manuscript shows good quality and is well written, supported by their data. However, the authors should consider the following:

  1. Introduction should also include why use CNT as filler, what advantages compared to other nanoparticles like ZnO.

In the Introduction of the revised manuscript, a brief additional text describing the advantages of using CNTs instead particles of ZnO as filler of nanocomposites prepared with the aim of being used as an antimicrobial material, was included.

  1. Maybe I’m missing, could author indicate what is the size of CNT like length and diameter. It will be better if the author can determine TEM  of CNT and f-CNT itself.

We appreciate and agree with the reviewer´s observation. However, the main objective of this work was to demonstrate the antimicrobial ability of the nanocomposites prepared with PHEMA chains grafted to functionalized CNTs instead of focusing on the dimensional characteristics of the CNTs. It is because of this reason why the sizes of the CNTs were not reported.

  1. The author should compare the impact of their studies against other toughening strategies in the literature

We appreciate the reviewer´s observation. In the Introduction section of the revised manuscript, this issue was addressed.

  1. The originality and novelties of the paper need to be further clarified.

The novelty of our study was included in the Introduction section of the revised manuscript.

Round 2

Reviewer 1 Report

The authors have answered all my comments and the paper can be accepted for publication. 

Minor editing of English language required.

Reviewer 2 Report

Accept in present form